# A Dynamic Load Modulation Power Amplifier with Ferroelectric-Based Tunable Matching Network

**DOI:** 10.3390/s24237571

**Published:** 2024-11-27

**Authors:** Pavel Turalchuk, Irina Filipiuk, Bayazet Iskakov

**Affiliations:** Department of Microelectronics and Radio Engineering, St. Petersburg Electrotechnical University “LETI”, 197022 St. Petersburg, Russia; iafilipiuk@etu.ru (I.F.);

**Keywords:** power amplifier, ferroelectric capacitor, energy efficiency, load modulation

## Abstract

Power amplifiers are crucial components that significantly influence the linearity and energy efficiency of next-generation communication system radio units. A key challenge in designing power amplifiers is managing high peak-to-average power ratio (PAPR) in order to achieve both high linearity and energy efficiency during back-off conditions. This paper presents simulation and measurement results for a dynamic load modulation power amplifier based on a ferroelectric tunable matching network to operate at 2.5 GHz. Experimental studies on a power amplifier with the tunable output matching network confirm its performance at 8 dB back-off while varying the control voltage applied to the ferroelectric element. Additionally, a bias modulator to adjust the transistor’s load in relation to input power was designed. Measurement studies of the dynamic load modulation power amplifier have demonstrated an efficiency of at least 50% at 8 dB back-off and more than 60% at peak power at 2.5 GHz. Furthermore, it was found that the modulator output voltage adjustment function on input power of the bias modulator affects the linearity of the output power. Different bias responses were studied and, as a result, the optimal output voltage response was found. The proposed load modulation power amplifier is promising for operation with high PAPR digital signals.

## 1. Introduction

The advancement of 5G wireless technology today demands increasingly stringent requirements for linearity, energy efficiency, and spectral efficiency in order to achieve high throughput. Modern communication systems leveraging orthogonal frequency-division multiplexing (OFDM) technology benefit from enhanced spectral efficiency. However, the use of numerous OFDM subcarriers leads to significant amplitude variations, causing the peak-to-average power ratio (PAPR) of OFDM signals to reach levels of 8–12 dB. A critical aspect of designing power amplifiers (PAs) is addressing this high PAPR to ensure both high linearity and energy efficiency across a wide dynamic range.

One of the most widely used solutions is the use of load-modulated and bias-modulated PA architecture. The latter, known as envelope tracking, involves tuning the power amplifier’s supply voltage in response to variations in the amplitude of the RF signal’s envelope. This approach enhances both the efficiency and linearity of the power amplifier. However, envelope-tracking PAs encounter challenges in designing the ET power supply, particularly as the bandwidth and PAPR of the envelope signal increase [1].

In the load-modulated technique, the load of the PA is modulated based on the input RF signal amplitude. Several fundamental load-modulated PA architectures exist, including the Doherty PA [2,3], load-modulated balanced power amplifiers (LMBA) [3,4], and amplifiers with dynamic load modulation (DLM) [5,6]. Typically, Doherty structures and LMBA use an additional peak stage amplifier that modulates the impedance of the main stage, providing high linearity and low power consumption in back-off. DLM is a single-stage structure that could be an addition of the main PA to extend the back-off range. The basic idea of dynamic load modulation is to tune the output matching network (OMN) to maintain high efficiency in back-off. It is known that OMN provides impedance transformation to the transistor output in order to provide optimal load to achieve higher efficiency. In order to modulate the transistor optimal load, a tunable OMN is utilized, for instance, a T-circuit based on varactors in [7], while the OMN proposed in [8] employs an OMN based on J-inverters. Additionally, ref. [9] discusses using a large number of varactors to reduce tunable capacitance while enhancing power-handling capabilities. Typically, varactor-based tunable matching networks are utilized to enhance the PA efficiency in back-off [10,11,12]. Also, integrated designs, such as integrated GaN diodes, could be beneficial for tunable MMIC PAs in terms of increasing the frequency of operation [13]. However, the nonlinear distortions introduced by these varactors can negatively affect the overall linearity of PA performance, which is crucially important for wireless applications. Typically, ferroelectric elements exhibit higher linearity, lower power consumption, and better power-handling capabilities than traditional semiconductor varactors [14,15]. In addition, the thin-film ferroelectric element has a switching speed of less than 1 ns and a quality factor of more than 100 MHz to 10 GHz [16]. These properties of the ferroelectric elements can potentially lead to high PA efficiency and high power operation. In the works [17,18], a load modulation of the PA load using a thick-film barium-strontium-titanate (BST) varactor was demonstrated to provide 6 dB back-off.

In this paper, an original design of a dynamic load modulation PA based on ferroelectric (FE) integrated circuit is proposed. This paper discusses the DLM PA design procedure with the proposed tunable OMN using the load-pull technique. Experimental studies of the PA with tunable OMN have been performed that confirm the 8 dB back-off with changing control voltage at the FE element. Finally, a bias modulator has also been designed to automatically modulate the load of the transistor with varying input signal strength. The measurement study of the DLM power amplifier demonstrated an efficiency of at least 50% in 8 dB back-off. In addition, it was observed that the linearity of the output power response is influenced by the modulator output voltage, which was investigated through experimental studies.

## 2. Dynamic Load Modulation Power Amplifier Design

### 2.1. PA with Extended Back-Off Design Procedure

The dynamic load modulation power amplifier is illustrated in Figure 1. The main concept of the DLM PA involves modulating the transistor load provided by the OMN using embedded tunable capacitors or semiconductor varactors. The parameters of the OMN are dynamically modified based on the amplitude of the input signal envelope to optimize the transistor load conditions for improved efficiency, output power, or linearity. Consequently, the control voltage applied to the tunable components must vary in sync with the signal envelope. Typically, the bias supply modulator generates this control voltage by amplifying a detected signal envelope coupled from the PA input. The proposed design for the OMN utilizes ferroelectric capacitors to achieve excellent power consumption, high power capability, and high linearity performance.

Typically, the transistor optimal load impedance that corresponds to the maximum efficiency or the output power is provided by the load-pull technique using a Smith chart. Here, a purely reactive load in the form of open-end stubs provides the condition for the transistor termination at the corresponding harmonic frequencies. The parameters of the harmonic matching circuit that allow the provision of higher efficiency are independent of the matching circuit parameters at the fundamental frequency, which simplifies the circuit synthesis. Choosing the output matching circuit parameters at the fundamental frequency allows the provision of a tradeoff between efficiency and output power [19].

At the same time, the PA efficiency could be adapted for the required output power by varying the optimal load resistance. For such purpose, using the load-pull technique, PA efficiency contours are simulated depending on the transistor load. Figure 2 presents the PA efficiency contours for a 6 W GaN HEMT transistor (Cree cgh40006 (Durham, NC, USA)) at 2.5 GHz center frequency in a Smith chart. It can be noted that the range of efficiency values from 65% to 50% corresponds to the different input power that leads to 8 dB output power back-off. In order to maintain high efficiency over the required back-off, a load tuning technique is used to adapt the load to an optimal value, with respect to the output power.

### 2.2. OMN Based on FE Tunable Capacitor

The OMN transforms a 50 Ohm load resistance (R_L_) into a complex conjugate impedance at the transistor’s output (*Z_out_*). In the case when the maximum efficiency contours shift depending on the input power, the load transformed with the OMN to the transistor’s output must also be positioned within the high-efficiency contour region. This allows high efficiency to be achieved in the back-off.

The OMN is represented by a T-type circuit, consisting of two transmission line sections and a shunt-tunable capacitor, as shown in Figure 2b. The parameter selection of the transmission line sections allows the provision of the transformation of the 50 Ohm impedance to the transistor optimal load, providing maximum efficiency at average power. The varying capacitance of the optimal load is tuned as the input power changes. The capacitance is tuned so that the impedance trajectory maintains high efficiency, depending on the input signal. According to the modeling results indicated by the dashed line on the Smith chart (Figure 2a), the impedance trajectory of the OMN optimally aligns with the high-efficiency contours. This trajectory is achieved by varying the capacitance between 0.8 and 2.35 pF.

A passive tunable integrated circuit (IC) TCP-4127UB, which utilizes a ferroelectric (FE) capacitor as its tunable element, was employed in this study. The IC includes an embedded DC biasing circuit. The frequency limitation is caused by the parasitic parameters of the capacitor electrodes and mainly by the embedded DC biasing circuit. To avoid frequency limitation for a specific IC, it is necessary to use a capacitor with a bias circuit designed for a required frequency range. This device operates within a frequency range of 700 MHz to 2.7 GHz, with a control voltage of up to 24 V, making it suitable for high-power amplifiers. The capacitor’s parameters were experimentally investigated, leading to the development of an equivalent model based on the extracted circuit parameters.

The de-embedding of the FE IC parameters was conducted using the TRL calibration technique. As a result, the measured frequency-dependent characteristics of FE capacitance at various control voltages, as illustrated in Figure 3a, was established. Additionally, Figure 3b displays the voltage-dependent characteristics of the FE capacitance at 2.5 GHz. As a result, the capacitance varied from 3.8 pF to 0.8 pF, corresponding to the control voltage variations from 2 V to 22 V.

Based on the extracted element parameters, an equivalent circuit for the capacitor was developed, as shown in Figure 2c. The values of the parasitic parameters (capacitance, inductance, and resistance) were estimated by fitting a simulated frequency-dependent response with measured data (see Figure 3). The shunt resistance Rq(f) = Q/(2π f C(f)) characterizes the quality factor of the component. The Q-factors were estimated to be about about 60 at 22 V applied voltage and 2.5 GHz central frequency. The insertion loss dependence of the OMN-based FE element on the capacitance value is calculated using extracted parameters. The insertion loss for different frequencies is shown in Figure 2c. Next, the developed equivalent circuit was used when modeling the characteristics of the load-modulated PA design.

### 2.3. Design of the PA with Tunable Matching Network

To confirm the calculation made in the previous section, the design of the power amplifier based on the CGH4006 transistor with tunable OMN was developed. The proposed design of the ferroelectric-based power amplifier is illustrated in Figure 4a. The output matching circuit includes a proposed T-network with a tunable FE capacitor. The OMN was designed to satisfy the trajectory of the impedance at the transistor’s output (*Z_out_*), as calculated previously (see Figure 3). The input matching circuit represents an impedance transformer with a transmission line section to provide matching with 50 Ohm input at large-signal operation. A laminate WL-CT338 (Shenzhen Bicheng Electronics Technology Corporation, Shenzhen, China) with a thickness of 0.508 mm and a dielectric constant of 3.38 and tanδ = 0.0029 (10 GHz) was used as the substrate. The gain and output power as functions of the input power results obtained by electrodynamic modeling (MoM) of the PA structure using a nonlinear transistor model are shown in Figure 4b. The modeling results of PA efficiency as a function of the output power for different values of the *C_FE_* are illustrated in Figure 4c. The large-signal simulation results of PA efficiency are obtained using harmonic balance methods. According to the modeling results, when varying the capacitance value in the range of 0.7–1.6 pF, the dynamic range (where the efficiency does not reduce below 50%) is 8 dB (30 dBm–38 dBm) at an operational frequency of 2.5 GHz.

### 2.4. Load-Modulated PA Measurement Results

The load-modulated PA has been fabricated and measured. A photograph of the fabricated ferroelectric-based load-modulated power amplifier prototype is shown in Figure 5a. A Vector Network Analyzer Rohde & Schwarz ZNB20 in the power sweep regime is utilized for PA output power measurement, while R & S Hameg hmp2020 is used for *DC* supply and *DC* power consumption measurement. The transistor drain supply is set to 24 V, and the gate voltage bias is −3.3 V, which corresponds to the B class of PA operation. The measured results of PA efficiency as a function of the output power at 2.5 GHz is shown in Figure 5b. According to the obtained results, the efficiency exceeds 50% in 8 dB back-off (30–38 dBm) while providing tuning of the OMN by applying control voltage between 2 and 16 V to the FE capacitor. It should be noted that, at 8 dB backoff (30 dBm output power), the efficiency of the load-modulated PA is about 20% higher. The absolute efficiency, however, is worse than the PA modeling results, due to higher losses in the test fixture. The measured frequency response of the drain efficiency and the output power for different voltage levels applied to the FE capacitor are presented in Figure 6. The most pronounced frequency dependence of the efficiency is observed at a voltage of less than 4 V, when the capacitance corresponds to its maximum value. With an efficiency of no worse than 40%, the frequency range is 4.3% (2.47–2.58 GHz) at average power. At the maximum output power, the efficiency varies within 50–75% within the 2.4–2.6 GHz frequency range. The output power unevenness versus frequency does not exceed 1 dB in the 2.4–2.6 GHz band. It should be noted that an increased bandwidth could be achieved by using multi-stage OMN with multiple capacitors but will lead to higher losses and reduced efficiency.

## 3. Dynamic Load Modulation PA with Control Bias Modulator

### 3.1. Capacitor Control with Bias Modulator

Typically, PA architecture utilizes additional bias voltage for the nonlinear element using a signal generator that is controlled by the level of input power [8,20]. In this paper, the operation of the PA in conjunction with a bias modulator is demonstrated. The control bias modulator regulates the control voltage applied to the FE capacitor, depending on the input signal power, with the aim of ensuring maximum efficiency operation of the power amplifier in the back-off. The envelope amplitude of the input signal is detected with a thru power detector ADL5906 (Analog Devices, Hongkong, China) through a directional coupler at the PA input. Since the range of the detected voltage from the amplitude detector is low, a differential stage using operational amplifiers is employed to amplify the envelope, including a buffer amplifier LM358 (National Semiconductor, Santa Clara, CA, USA) and a high-speed amplifier LM6172 (Texas Instruments, Hongkong, China). A scheme for the bias modulator is illustrated in Figure 7a. The output modulator voltage *U_cap_* is evaluated in the following way:(1)$$U_{cap}\left(U_{d}\right)=k\left(U_{d}-U_{b}\right),$$
where $k=\frac{R_{3}}{R_{1}}$; *U_d_*—detected input signal envelope; and *U_b_*—introduced bias voltage for differential stage to provide voltage output in the range of 2–22 V. Adjusting the bias level *U_b_* of the buffer amplifier allows the setting of the voltage gain slope characteristic, as shown in Figure 7b. Changing the voltage gain slope characteristic allows the control of the linearity of the PA output power while maintaining the efficiency response.

### 3.2. Measurement Results of the DLM PA with Capacitor Bias Modulator

The fabricated power amplifier based on an FE capacitor was experimentally studied using the bias modulator. The bias modulator regulates the control voltage applied to the FE capacitor, depending on the input signal power, correspondingly to the voltage regulation response shown in Figure 7. The DLM PA drain efficiency was experimentally studied as a function of the output power, for which the results are shown in Figure 8. According to the presented results, it can be seen that the drain efficiency of the DLM PA agrees well with the drain efficiency curves shown in Figure 6b, which correspond to the envelope of the efficiency curves obtained at different control voltages. The efficiency is also improved by 20% with PA dynamic load modulation compared to a fixed transistor load in the case of 8 dB back-off. It should be stressed that the DC power consumption of the envelope modulator does not take into account in the evaluation of PA efficiency. The DC power consumption of the modulator is about 0.3 W and can be reduced by another schematic solution. This DC power consumption results in a power amplifier efficiency reduction of up to 5% at average output power (30 dBm) and below 1% at peak output power (38 dBm).

The linearity of the gain and output power response, with respect to the input power, depends on the slope of the voltage adjustment characteristic. Therefore, different voltage supply adjustment profiles are studied to demonstrate the influence on the gain and the output power response. The measured results of the gain and the output power versus the input power at different slopes of the voltage adjustment characteristics are shown in Figure 9a. In accordance with the presented results, voltage adjustment profiles III and IV provide the best result in terms of the linearity of the gain and output power, increasing the input power and improved efficiency. It should be stressed that the dependence of efficiency on output power does not change markedly when the slope of the voltage adjustment characteristic of the modulator changes, as shown in Figure 9b. In the case of IV voltage response, the efficiency has a more pronounced S-shaped character, in which, in the small-signal regime, the efficiency decreases. As a rule, nonlinear gain response can affect the deterioration of PA linearity when operating with modulated signals. If the gain response is more linear, then it could potentially considerably simplify the linearization methodology, with respect to nonlinear behavior [21]. Hence, one can expect better linearity of the PA characteristics when using the proposed PA with properly selected modulator supply voltage shaping.

In Table 1, the performance of the proposed PA in this work is compared to other DLM PAs. All PAs are designed using GaN transistors but using different varactor technologies. The performance of the proposed device is similar to that of other PAs based on BST ferroelectric technology but has a larger back-off.

## 4. Conclusions

The design of a 6 W power amplifier featuring dynamic load modulation at 2.5 GHz is proposed herein. A passive ferroelectric-based output matching network was designed to optimize the transistor’s load impedance as the output power varies. The power amplifier was EM-simulated using an equivalent circuit model of the ferroelectric capacitor. The parameters of the transistor’s output matching network were evaluated using load-pull techniques to ensure high efficiency during back-off conditions. The load-modulated power amplifier has been fabricated and tested, with measurements indicating that efficiency exceeds 50% at an 8 dB back-off while tuning the output matching network by applying control voltages ranging from 2 to 16 V. Additionally, the control bias modulator used to adjust the voltage applied to the capacitor based on the input signal power was developed. The experimental studies of the designed power amplifier using the bias modulator aimed to maximize efficiency during back-off. The results demonstrate a 20% improvement in efficiency with dynamic load modulation at 8 dB back-off, and it was shown that the linearity of the output power is influenced by the slope of the voltage adjustment characteristic of the bias modulator. It should be noted that the DLM is of practical interest as a single PA stage, and the main amplifier stages in Doherty or LMBA structure can be supplemented with load modulation techniques based on tunable FE-based OMN to improve the efficiency of the back-off.

## Figures and Tables

**Figure 1 sensors-24-07571-f001:**
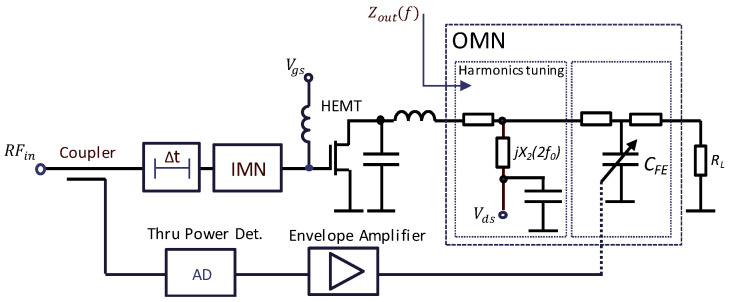
The scheme for DLM PA based on ferroelectric tunable matching network.

**Figure 2 sensors-24-07571-f002:**
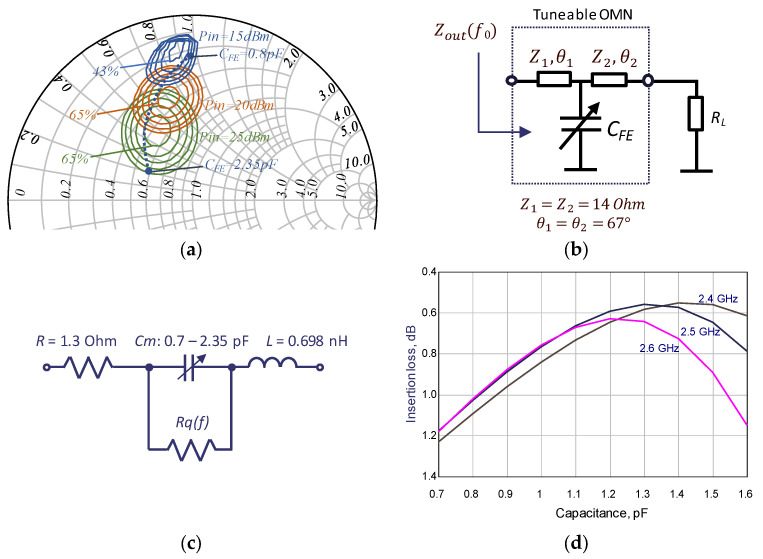
(**a**) The results of load-pull for a GaN HEMT transistor at 2.5 GHz central frequency. The dashed line—the impedance *Z_out_* provided by tunable OMN with changing *C_FE_* in the range of 0.8–2.35 pF; (**b**) the design of the output matching circuit; (**c**) the FE IC equivalent circuit with the extracted parameters; (**d**) the insertion loss of the OMN-based FE element versus capacitance value for different frequencies.

**Figure 3 sensors-24-07571-f003:**
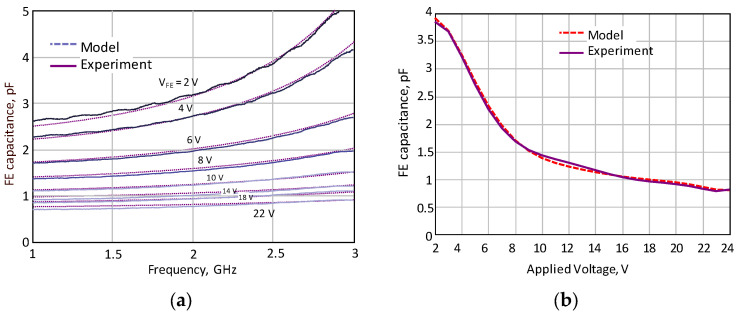
(**a**) The modeling and measurement results of the frequency dependence of the FE capacitance for different applied voltages; (**b**) Measurement of the FE capacitance as a function of applied voltage at 2.5 GHz compared with the modeling results.

**Figure 4 sensors-24-07571-f004:**
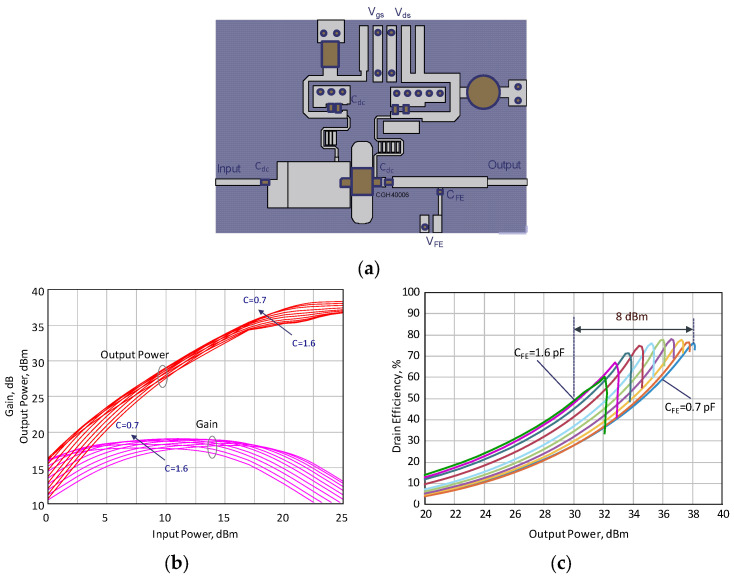
(**a**) The design of the PA with FE-based OMN; (**b**) modeling results of the PA gain and output power versus the input power for changing values of C_FE_; (**c**) modeling results of the PA efficiency as a function of the output power for different values of C_FE_ (from 1.6 pF to 0.7 pF).

**Figure 5 sensors-24-07571-f005:**
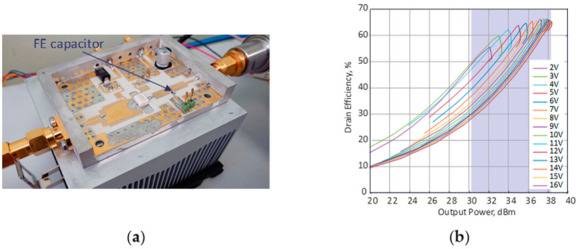
(**a**) Fabricated load-modulated PA with ferroelectric-based OMN; (**b**) Measured drain efficiency versus the output power for different values of the voltage applied to the FE varactor at 2.5 GHz.

**Figure 6 sensors-24-07571-f006:**
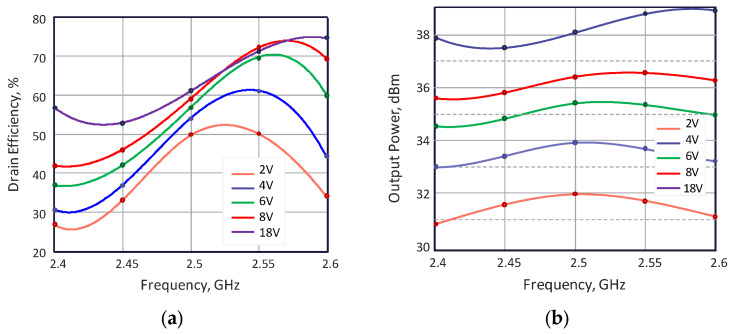
(**a**) Measured frequency response of the PA efficiency and (**b**) the output power for different levels of voltage applied to the FE capacitor.

**Figure 7 sensors-24-07571-f007:**
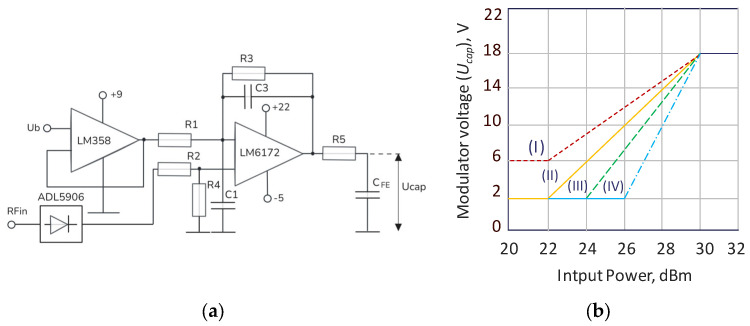
(a) Scheme for the bias modulator; (b) output voltage adjustment characteristics of the bias modulator as a function of the input power.

**Figure 8 sensors-24-07571-f008:**
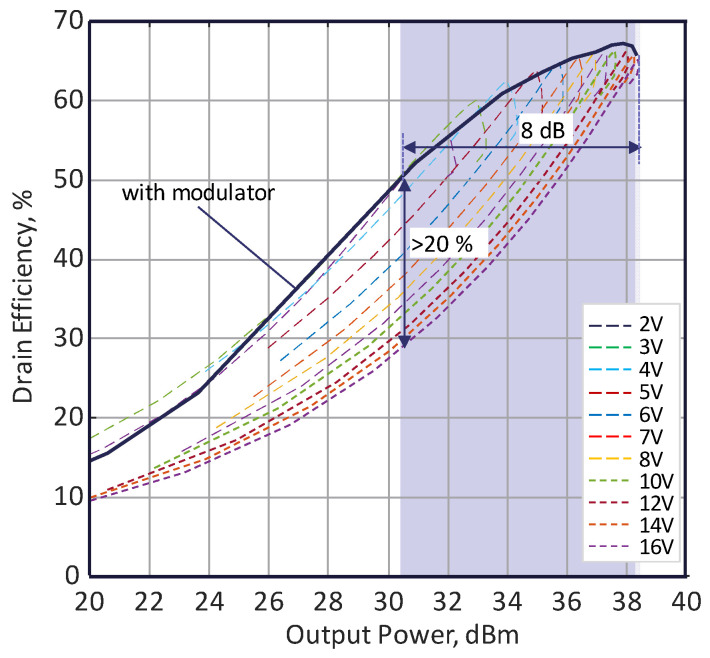
Measured drain efficiency versus the output power of the DLM PA with a bias modulator.

**Figure 9 sensors-24-07571-f009:**
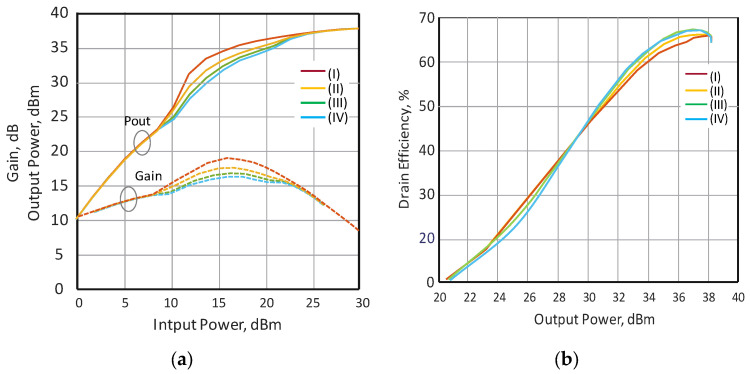
(**a**) Measured response of the gain and output power of the DLM PA as a function of the input power for different voltage adjustment responses; (**b**) measured efficiency of the DLM PA versus the output power for different voltage adjustment profiles.

**Table 1 sensors-24-07571-t001:** Characteristics comparison with other published DLM PAs.

Peak Power (Transistor Type)	Varactor Technology	Freq, GHz	BO, dB	Eff, % (Peak Power/BO)	Ref.
38 dBm (GaN)	SiC	2.08	8	60/45	[12]
41 dBm (GaN)	SiC	1.80–2.20	6	50/50	[18]
39 dBm (GaN)	LDMOS FET	2.65	5	70/55	[11]
43 dBm (GaN)	BST	1.65–1.95	6	70/57	[17]
45 dBm (GaN)	BST	1.85	6	55/29	[15]
38 dBm (GaN)	BST	2.5	8	65/50	This work

## Data Availability

Data are contained within the article.

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
