# Peer review of "A Dynamic Load Modulation Power Amplifier with Ferroelectric-Based Tunable Matching Network"

_sensors, 2024, doi:10.3390/s24237571_

Round 1
Reviewer 1 Report
Comments and Suggestions for Authors
1. What is the definition of Pin in the PAE calculation? Is it the input of the main path of the PA, or the total input, RF_in? In other words, is Pin the power before the coupler or the power after the coupler?
2. In the selection of reconfigurable devices, why choose Fe capacitors over ordinary varactors and PIN diodes? Please explain the reasons for choosing Fe capacitors.
3. According to Figure 3(a), for V_FE = 2V and 4V, the FE capacitance exhibits significant fluctuations around the center frequency of 2.5 GHz. Will such large fluctuations within the bandwidth have a significant impact on the amplified signal when the designed PA amplifies a wideband signal?
4. In Figure 4, are the values of each parameter marked incorrectly? For example, what does "7÷2" mean? Does "R=1, 3 ohm" indicate R = 1 ohm or 3 ohm?
5. Please standardize the notation. For example, in Section 2.3, where the term "power-added efficiency" is defined as the abbreviation "PAE", please use "PAE" consistently throughout the rest of the text. There are also other instances of inconsistent abbreviation usage in the text. Please have the author correct them.
6. Please provide a comparison between the proposed structure and research in similar directions, particularly in comparison with the latest research and the most classic studies.
Author Response
Thank you very much for taking the time to review this manuscript. Please find the detailed responses below and the corresponding corrections highlighted in the paper. I hope the revised manuscript became better with taking into account reviewer comments.
Point-by-point response to Comments and Suggestions for Authors
Comments 1: What is the definition of Pin in the PAE calculation? Is it the input of the main path of the PA, or the total input, RF_in? In other words, is Pin the power before the coupler or the power after the coupler?
Response 1: Thank you for pointing this out. Typically, a coupler with a 20 dB coupling is used, so there is no significant difference in PAE if it is determined before or after the directional coupler. But, we decided that it is better to give the PA efficiency or drain efficiency for all results obtained from simulation and experiment instead of PAE. Corresponding changes made to the paper: Fig.5 (b) is modified and in the text. The corresponding corrections highlighted in the paper.
Comments 2: In the selection of reconfigurable devices, why choose Fe capacitors over ordinary varactors and PIN diodes? Please explain the reasons for choosing Fe capacitors.
Response 2: It is promising that properties of the ferroelectric elements can potentially leads to the PA high efficiency and high power operation. Typically, ferroelectric elements exhibits high linearity, lower power consumption and excellent power-handling capabilities than traditional semiconductor varactors. Intermodulation characteristics (IM3) also exhibits better performance with respect to semiconductor elements. So, we expect low nonlinearity in the PA with using ferroelectric varactors. These reasons are provided in the introduction and highlighted.
Comments 3: According to Figure 3(a), for V_FE = 2V and 4V, the FE capacitance exhibits significant fluctuations around the center frequency of 2.5 GHz. Will such large fluctuations within the bandwidth have a significant impact on the amplified signal when the designed PA amplifies a wideband signal?
Response 3: We agree with this comment. The capacitance has markable frequency dependence. Therefore, we include in the paper the frequency response of the efficiency and the output power, Figure 6. Also, some comments in the text are provided and highlighted.
Comments 4: In Figure 4, are the values of each parameter marked incorrectly? For example, what does "7÷2" mean? Does "R=1, 3 ohm" indicate R = 1 ohm or 3 ohm?
Response 4: Thank you for pointing this out. Agree. The parameters are revised.
Comments 5: Please standardize the notation. For example, in Section 2.3, where the term "power-added efficiency" is defined as the abbreviation "PAE", please use "PAE" consistently throughout the rest of the text. There are also other instances of inconsistent abbreviation usage in the text. Please have the author correct them.
Response 5: Agree. The text is revised with respect to reviewer comments.
Comments 6: Please provide a comparison between the proposed structure and research in similar directions, particularly in comparison with the latest research and the most classic studies.
Response 6: Thank you for pointing this out. The table (Table 1) is added to the paper to provide a comparison between the proposed structure and published research papers.

Reviewer 2 Report
Comments and Suggestions for Authors
This paper proposed a dynamic load modulation power amplifier withFerroelectic Based Tunable Matching Network. Here are my comments.
1. The abstract requires grammatical revision to ensure consistency in singular and plural forms, such as changing "A power amplifiers" to "A power amplifier". Additionally, ensure proper capitalization, such as using lowercase for "peak-to-average power ratio".
2. As the capacitance of the ferroelctric (FE) capacitor varies, the Q-factor of CFE will change, impacting the output matching loss. It would be beneficial to present a plot showing OMN losses as a function of CFE to illustrate this effect.
3. The parasitic parameter value in Fig. 4 appear to contain typographical errors, such as R=1,3 ohms, Cm: 0,7%2, 35 pF, and L=0,698nH. Please verify and correct these values in Fig. 4
4. The printed circuit board (PCB) utilized a WL-CT338 laminate with a thickness of 0.508 mm. Please provide the transmission line loss or attenuation constant for the WL-CT338 substrate to enhance the technical understanding
5. Fig.9 (a) shows the output power vs. input power, with gain being less 0 around 20 dBm and increasing at 23 ~ 24 dBm. Please include a plot of gain versus input power to identify the 1-dB compression point and discuss the result.
6. The manuscript lacks Section 4, as it jumps from Section3: Dynamic Load Modulation PA with Control Bias Modulator to Section 5: Conclusion. Please include a section 4 or modify the section number and add a comparison table with relevant references to provide a comprehensive analysis.
Author Response
Thank you very much for taking the time to review this manuscript. Please find the detailed responses below and the corresponding corrections highlighted in the paper. I hope by introducing these corrections with taking into account reviewer comments the paper became better.
Point-by-point response to Comments and Suggestions for Authors
Comments 1: The abstract requires grammatical revision to ensure consistency in singular and plural forms, such as changing "A power amplifiers" to "A power amplifier". Additionally, ensure proper capitalization, such as using lowercase for "peak-to-average power ratio".
Response 1: Thanks. The text is revised with respect to reviewer comments. The corrections highlighted in the paper.
Comments 2: As the capacitance of the ferroelctric (FE) capacitor varies, the Q-factor of CFE will change, impacting the output matching loss. It would be beneficial to present a plot showing OMN losses as a function of CFE to illustrate this effect.
Response 2: Thank you for important comment. Actually, the output matching loss depend on CFE. So, we add to the paper the plot showing the insertion loss of the OMN based FE element versus capacitance value for different frequencies – Fig.2(c). Also, some comments in the text are provided and highlighted.
Comments 3: The parasitic parameter value in Fig. 4 appear to contain typographical errors, such as R=1,3 ohms, Cm: 0,7%2, 35 pF, and L=0,698nH. Please verify and correct these values in Fig. 4
Response 3: Thank you for pointing this out. The parameters are corrected.
Comments 4: The printed circuit board (PCB) utilized a WL-CT338 laminate with a thickness of 0.508 mm. Please provide the transmission line loss or attenuation constant for the WL-CT338 substrate to enhance the technical understanding
Response 4: Agree with this comment. The parameters are added to text.
Comments 5: Fig.9 (a) shows the output power vs. input power, with gain being less 0 around 20 dBm and increasing at 23 ~ 24 dBm. Please include a plot of gain versus input power to identify the 1-dB compression point and discuss the result.
Response 5: Thank you a lot for pointing this out. Results were incorrectly presented. The Fig.9(a) showing the output power vs. input power is revised and also gain is added to the plot. Additionally, modeling results of the PA gain and output power versus the input power for changing values of the CFE are introduced in Fig.4(b).
Comments 6: The manuscript lacks Section 4, as it jumps from Section3: Dynamic Load Modulation PA with Control Bias Modulator to Section 5: Conclusion. Please include a section 4 or modify the section number and add a comparison table with relevant references to provide a comprehensive analysis.
Response 6: Thank you for pointing this out. Section number is corrected. Also, the table (Table 1) is added to the paper to provide a comparison between the proposed structure and published research.

Reviewer 3 Report
Comments and Suggestions for Authors
This work presents the implementation of a DLM PA at 2.5 GHz. However, a critical missing aspect is the lack of experimental results with modulated signals, which are essential to validate the concept. It appears that only a static characterization of the performance is provided, making it unclear how the authors conclude that shaping profile III offers better linearity without testing it. In addition, the authors should consider the following points.
The effectiveness of DLM on PAs strongly depends on the operating frequency. Therefore, it is essential that the authors mention the operating frequency (2.5 GHz) in the abstract and discuss its relevance in the introduction. Some DLM solutions support frequencies up to 6 GHz, while higher frequencies require MMIC implementation, where varactor technology is still under development (e.g., S. Cangini et al., "Evaluation of Integrated GaN Diodes as Varactors for Tunable MMIC PAs from C- to K-Band," 2024 19th European Microwave Integrated Circuits Conference (EuMIC), Paris, France, 2024, pp. 14-17).
What are the frequency limitations of the ferroelectric elements used in this design? The component used here appears to operate up to 2.7 GHz. Please include a discussion on the state-of-the-art in terms of available technology and performance for these elements. Additionally, the authors should comment on the insertion loss, which can significantly impact the PA's PAE.
Could the authors better explain the chosen topology for the OMN (Fig. 2b) and clarify why the variable capacitor is placed there?
A performance comparison table with other PA solutions at this operating frequency should be included. Additionally, the efficiency results should account for the power consumption of the envelope modulator.
Author Response
Thank you very much for taking the time to review this manuscript. Please find the detailed responses below and the corresponding corrections highlighted in the paper.
Point-by-point response to Comments and Suggestions for Authors
Comments 1: This work presents the implementation of a DLM PA at 2.5 GHz. However, a critical missing aspect is the lack of experimental results with modulated signals, which are essential to validate the concept. It appears that only a static characterization of the performance is provided, making it unclear how the authors conclude that shaping profile III offers better linearity without testing it. In addition, the authors should consider the following points.
Response 1: We absolutely agree with this statement. But, at the current stage of experimental research, it is not possible to carry out the study PA nonlinearity using modulated signals. Such work is underway and we will present these results in the future. The point is that it is necessary to align the signals in time/phase in the arms of the main PA and modulator so that the control is synchronized with the modulated signal. At the same time, the nonlinearity of the output power or gain characteristics affects the modulated signal with some bandwidth. The gain/pout response with linear behavior leads potentially to more simplified linearization technique (DPD) with respect to non-linear response. Therefore, we believe that at this stage it is important to pay attention to the linearity of these static characteristics depending on the modulator parameters. So, we add some comments to the text and reference to prove importance of the gain/pout linear response.
Comments 2: The effectiveness of DLM on PAs strongly depends on the operating frequency. Therefore, it is essential that the authors mention the operating frequency (2.5 GHz) in the abstract and discuss its relevance in the introduction. Some DLM solutions support frequencies up to 6 GHz, while higher frequencies require MMIC implementation, where varactor technology is still under development (e.g., S. Cangini et al., "Evaluation of Integrated GaN Diodes as Varactors for Tunable MMIC PAs from C- to K-Band," 2024 19th European Microwave Integrated Circuits Conference (EuMIC), Paris, France, 2024, pp. 14-17).
Response 2: Thank you a lot for pointing this out. We include a discussion in the introduction. Some corrections are done in the text and highlighted.
Comments 3: What are the frequency limitations of the ferroelectric elements used in this design? The component used here appears to operate up to 2.7 GHz. Please include a discussion on the state-of-the-art in terms of available technology and performance for these elements. Additionally, the authors should comment on the insertion loss, which can significantly impact the PA's PAE.
Response 3: Thank you a lot for your comment, as frequency characteristics is of interest for readers. The frequency limitation is caused by the parasitic parameters of the capacitor electrodes and mainly by the embedded DC biasing circuit. To avoid frequency limitation for a specific IC, it is necessary to use a SMD capacitor with a bias circuit designed for a required frequency range. Also, we add additional plots showing insertion loss of the matching network and the frequency response of the efficiency and the output power (Fig. 6). Also, some comments in the text are provided and highlighted.
Comments 4: Could the authors better explain the chosen topology for the OMN (Fig. 2b) and clarify why the variable capacitor is placed there?
Response 4: The OMN is represented by a T-type circuit, consisting of two transmission line sections and a shunt tunable capacitor, as shown in Fig. 2(b). The parameters selection of the transmission line sections allows providing transformation of the 50 Ohm impedance to the transistor optimal load, providing maximum efficiency at average power. Varying the capacitance the optimal load is tuned as the input power changes. We use classical representation of the matching circuit by using “T”, “Π”, “Γ” – type to provide matching with transistor complex impedance. The T-type circuit takes single capacitor and the network is symmetrical.
Comments 5: A performance comparison table with other PA solutions at this operating frequency should be included. Additionally, the efficiency results should account for the power consumption of the envelope modulator.
Response 5: Thank you for pointing this out. The table (Table 1) is added to the paper to provide a comparison between the proposed structure and published research.
Regarding to for the power consumption of the envelope modulator: the dc power consumption of the modulator does not taken into account during evaluating the PA efficiency. The dc power consumption of the modulator is estimated about 0.3 W, but can be reduced by another schematic solution. This DC power consumption results in a power amplifier efficiency reduction of up to 5% at average output power (30 dBm) and below 1% at peak output power (38 dBm). We add this comment to the text.
Thanks to the reviewer’s a lot for valuable comments.

Round 2
Reviewer 1 Report
Comments and Suggestions for Authors
Thanks for the response, I only have one small comments: Is the labeling “cm: 0.7 ÷ 2.35pF” in Figure 2(a) correct, and did the author intend to indicate “cm: 0.7 - 2.35pF”?
Author Response
Thank you very much for taking the time to review this manuscript.
Reviewer 2 Report
Comments and Suggestions for Authors
The authors well addressed the issues that reviewers were concerned.
Author Response

(The authors gave the same response as above.)

Reviewer 3 Report
Comments and Suggestions for Authors
The absence of the experimental results is the main limiting aspect here. Other than that, the authors have responded to the questions.
Author Response

(The authors gave the same response as above.)
